# Association of COVID-19 stimulus receipt and spending with family health

**Emma M. Reese**[1]*, **Noah Lines**[2], **Evan L. Thacker**[1], **Michael D. Barnes**[1]

**1** Department of Public Health, Brigham Young University, Provo, Utah, United States of America,
**2** Department of Philosophy, Brigham Young University, Provo, Utah, United States of America

* emmasheranian@gmail.com

## Abstract

In this study, we aimed to determine the impact of U.S. government stimulus payments on family health during the COVID-19 pandemic. We hypothesized that receiving stimulus checks is associated with better family health and the effect of stimulus check receipt differs by income level. Additionally, we hypothesized that spending on immediate needs and paying off loans is associated with worse family health, and the effects of this spending differ by income level. Participants included 456 registered Amazon Mechanical Turk (mTurk) users, stratified by income, marital status, and parental status. We used the Family Health Scale – Long Form to measure family health constructs: social-emotional health, healthy lifestyle, health resources, and social support. For all statistical analyses, we used SAS Studio 3.8. We performed an exploratory factor analysis to determine six spending profiles: loans, savings, housing, household supplies, durable goods, and medical costs. After adjustment, our multiple linear regression model found that mean family health and social-emotional health scores were higher among individuals who received all three checks, but this did not differ by income category. Mean family health and social-emotional health were lower among individuals who spent more significant portions of their stimulus checks on housing, household supplies, and medical costs. Spending greater portions of checks on medical costs was associated with lower scores among every family health construct except family healthy lifestyle. Among mid-to-high-income participants, family health scores were significantly lower, with more spending on housing, household supplies, durable goods, and medical costs, with similar results in the subscale scores. The reduction of family health scores with spending on medical costs and durable goods were more pronounced among the mid-to-high-income group than the low-income group. Stimulus payments may be a promising family policy method for improving overall family health; however, more research should address the differences between income groups and government assistance.

**Data availability statement:** All relevant data are within the paper and its Supporting information files.

**Funding:** This research received no external funding. Michael D. Barnes received internal funding from Brigham Young University Department of Public Health. The funders had no role in study design, data collection and analysis, decision to publish, or preparation of the manuscript.

**Competing interests:** The authors have declared that no competing interests exist.

## Introduction

In response to the early stages of the COVID-19 pandemic in 2020, the U.S. government passed legislation to minimize health and economic damage of the spreading virus and shelter-in-place policies, while encouraging spending. This legislation included three economic impact payments (stimulus checks) paid directly to qualified taxpayers [1]. The first payment, CARES Act, was made available beginning in spring 2020, with most households receiving $1,200. The second round, Consolidated Appropriations Act, provided up to $600 per adult and $600 per child using lower maximum income thresholds for qualified payments available in late 2020 and early 2021. The third, American Rescue Plan, provided $1400 per adult and $1400 per child using even lower maximum income thresholds for qualified payments beginning spring 2021 [2].

Significant financial stressors for families and individuals, such as economic downturns, compromised employment, and restrictions in purchasing and access to essential services, indicate the importance of economic spending [3]. While the pandemic has amplified these stressors for most people, minority families and lower-income populations have been more significantly impacted by the pandemic than those from higher socioeconomic status (SES) and race majorities [3]. For example, during peak pandemic shutdowns, employees of lower-income jobs such as food service were at a higher risk of exposure to COVID-19. In contrast, higher-income families typically worked safely from home [4]. Further, workers in lower-income families had higher unemployment or underemployment rates than higher-income families because many could not work from home [5,6].

Family health has been defined as "a resource at the level of the family unit that develops from the intersection of the health of each family member, their interactions and capacities, as well as the family's physical, social, emotional, economic, and medical care" [7]. Family functioning is significantly impacted by the COVID-19 pandemic, including social and emotional processes, lifestyle, resources, and social support [8]. The family disruptions caused by the pandemic affecting overall family health and well-being varied according to gender in rural and urban settings [9]. Most U.S. research points to economic and employment constraints [10–12], interruptions to family routines [13], reduced quality of life and family well-being [8,14], psychological distress [8,15–18], loneliness [19], and reduced access to important family resources such as medical care and essential community services [20]. In ideal conditions, family or household members would help to reduce feelings of stress, social isolation, and insecurity during times when normal activities or routines are disrupted.

The interconnectedness of family members is central to family systems theory, which states that the family unit is a social system with subsystems in which family members are interconnected and greatly affect each other's well-being and the family unit [21,22]. Family systems theory acknowledges that a family's reaction to stress influences its capacity to maintain or re-establish stability in the face of distress [22]. During the pandemic, researchers found that COVID-19 stressors within families predicted greater family discord – stressors predicted negative parenting techniques and parental conflict, which increased child distress and decreased family cohesion

[22,23]. Many of the early pandemic-related stressors experienced by families were economically based like losses of jobs or income. Thus, the economic impact payments provide an opportunity to explore the impact of household income and spending choices resulting from stimulus checks and their effect on family health status during the pandemic.

## Purpose

In this study, we aimed to determine the impact of government-assisted stimulus payments on family health and well-being amidst the stressors brought on by the COVID-19 pandemic in the U.S. We specifically ask, "How have stimulus payments and reported spending affected family health given the stressors of the COVID-19 pandemic? How is family health impacted by the interaction of income level with receiving and spending the stimulus payments?" Thus, we hypothesized that receiving stimulus checks is associated with better family health and the effect of stimulus check receipt differs by income level. Additionally, we hypothesized that spending on immediate needs and paying off loans is associated with worse family health, and the effects of this spending differ by income level.

## Materials and methods

### Participants and sampling

From June to July 2021, 650 registered Amazon Mechanical Turk (mTurk) users participated in the survey. MTurk is a crowdsourcing marketplace that businesses and individuals can use to outsource data collection and processing for various purposes, including research. MTurk users provide a sample more diverse than typical convenience samples, and virtually complete tasks requested by researchers, such as surveys, data validation, and more. The mTurk users receive an incentive, typically financial compensation, for their work [24]. Eligibility for the survey was restricted to those who had at least a 95% Human Intelligence Task (HIT) rating. Research involving the most reputable mTurk samples suggests oversampling up to 30% to create a margin of clean and valid data responses, known as active response rate [25]. The 456-participant sample size would allow sufficient power based on the anticipated number of factors for the various economic items and family health status scale [26]. Participants were removed during data cleaning if <80% of the survey was completed, if survey participants completed the survey in less than 40% of the average time to take the survey, and failure to pass "lie detection" questions, resulting in a sample size of 456. They were at least 18 years of age and resided in the United States. We specified the following mTurk selection criteria to enlist a sample of U.S. family households of which 15% would have income < $25,000, 40% would be parents, and 20% would be married. Survey respondents received $2.00 after completing the survey. Approval for the use of human subjects was given by the Brigham Young University Institutional Review Board Human Research Protection Program.

### Survey administration procedures

All surveys were estimated to take 10 minutes to complete. A link to the survey was posted on mTurk. Qualifying participants saw the posting on mTurk that described the survey and the approximate length of time. If potential participants agreed to participate, they then accepted the HIT offered through mTurk and were directed to the Qualtrics survey page. The first question on the survey was a consent form. Participants checked the box signifying their agreement with the consent (in lieu of a signature) before seeing the survey questions. After completing the survey, they received a code which they entered into Amazon mTurk demonstrating completion of the HIT. This allowed them to be paid for their participation. Participants were paid through their Amazon mTurk worker account.

### Data collection materials

The survey consisted of a validated instrument, stimulus check questions, and demographic questions. The instrument included in the current study was the Family Health Scale to measure family health constructs [27].

**Family Health Scale – Long Form (FHS-LF).** The FHS-LF is a 32-item scale created to measure family health constructs with a family systems lens [27]. Response options for each item were recorded on a 5-point Likert scale ranging from *Strongly agree* to *Strongly disagree*. Negatively worded items were reverse coded so higher scores indicated better family health. The FHS-LF items were summed to create a cumulative score ranging from 0 to 128. Four subscales were identified through exploratory factor analysis as family social-emotional processes, healthy lifestyle, health resources, and external social support [27]. Social-emotional health represents internal familial processes such as emotional safety, connection, communication, satisfaction, and coping and has a cumulative score ranging from 0 to 52. Healthy lifestyle represents internal familial processes that address healthy behaviors and habits, and the cumulative score ranged from 0 to 24. Next, health resources represent health characteristics such as internal and external resources with a score ranging from 0 to 36. Finally, family external social support represents social support a family has and the score ranges from 0 to 16 [27].

**Stimulus check and spending variables.** Participants were asked whether they received each of the first, second, and third stimulus checks. For each check, if they answered *Yes,* the survey prompted them with further questions about what portion of the check (most, some, none) was spent on each of the following six categories: loans, savings, housing, household supplies, durable goods, and medical costs. For analyses based on the number of checks received as an independent variable, we dichotomized the responses to "three checks" versus "fewer than three checks" because of relatively small sample sizes for groups that reported receiving only two, only one, or no stimulus checks.

**Demographics.** Age, SES, and household size were measured as continuous variables. SES was measured using the MacArthur Ladder Scale of Subjective Social Status, wherein participants ranked themselves from 1 to 10, having the least money, education, and respected job (1) to the most money, education, and respected job (10). Higher rankings were considered higher SES [28]. Gender (male, female, or self-identified), education (less than high school, high school graduate, some college, two-year degree, four-year degree, master's degree, and professional or doctoral degree), race (White/Caucasian, Black/African American, Asian, Hispanic/Latino, Pacific Islander, Native American, Multiracial, and other), relationship status (cohabiting relationship, married, or other), and income level were measured as categorical variables. Income level categories were low (<$40,000), middle ($40,000 to $140,000), and high ($140,000+) income. The low-income category threshold of <$40,000 to define categories for this analysis was different from the aforementioned threshold of <$25,000 that was used for the mTurk sampling. This low-income threshold for analysis was based on PEW's American Trends Panel methodology [29] calculating the low-income tier for families as $39,800. Middle income was defined as $40,000 to $140,000 and high-income families above $140,000 [29–31]. Due to a low sample size of high-income individuals, we combined middle- and high-income categories to create a binary income variable (low-income or mid-to-high income). Employment status was also measured as a binary variable.

## Data Analysis

**Spending patterns determination.** Six spending types across three stimulus checks resulted in 18 spending variables. Correlations within a given spending type across three checks were higher than correlations across six spending types within a given check (S1 Table). To reduce the 18 variables into a smaller number of factors that would indicate spending profiles for analysis, we used principal component analysis with varimax rotation. Preliminary analyses suggested that the same spending type across the three stimulus checks would tend to load highly on the same factor. Therefore, to maximize interpretability of each factor as emphasizing spending of one type, we had an a priori preference to retain six factors for further analysis, one factor per spending type. To evaluate a six-factor solution relative to solutions retaining a larger or smaller number of factors, we considered holistically several criteria, including overall interpretability, eigenvalues, scree plots, total amount of variance accounted for by the set of factors, proportion of variance accounted for by each factor, and item communality. We did not rely solely on the eigenvalue-greater-than-one rule, as a strict application of that rule may lead to omitting a conceptually important factor that has an eigenvalue just below 1.0. With a six-factor solution, we found that

the highest item loadings in each factor corresponded to the same spending type across the three stimulus checks, which optimized overall interpretability (Table 1). Eigenvalues for the first five factors exceeded 1.0, and the Eigenvalue for the sixth factor (spending on medical care) was 0.9, indicating that the sixth factor would explain less variance than any individual variable. However, the scree plot suggested that factors one through four each accounted for a large degree of variance, factor five accounted for a small degree, factors six and seven each accounted for a moderate degree of additional variance, and factors eight and above each accounted for only a small degree of additional variance, therefore, retaining between four and seven factors appeared reasonable. Six factors together accounted for 80% of the standardized variance, compared with 76% of the standardized variance with five factors. After varimax rotation, variance explained by each factor ranged from 2.2 to 2.6, summing to 14.5 out of the total variance of 18. Item communality estimates ranged from 0.67 to 0.88, indicating that the six factors together explained a high proportion of the variance in each of the 18 original variables. We found that a solution retaining seven factors reduced overall interpretability because it resulted in high factor loadings for durable goods spending items being split across two factors, and a solution retaining five factors reduced overall interpretability because it resulted in high factor loadings for medical spending items being combined with high factor loadings for household supplies items in a single factor. Therefore, based on evaluating all those criteria, with a priority of achieving data reduction while optimizing conceptual interpretability, we settled on the six factor solution and retained six factors for subsequent analyses. This principal component analysis was conducted in a sample of 369 participants who received all three stimulus checks and answered all the spending items. To complete this principal component analysis, we used statistical software package SAS Studio 3.8.

**Table 1. Principal component analysis of spending variables.**

| | Six factors, each representing one spending type | | | | | | |
|---|---|---|---|---|---|---|---|
| | Loans | Savings | Housing | Household supplies | Durable goods | Medical costs | Communality (Total = 14.46) |
| Eigenvalue of the factor | 6.3 | 2.8 | 2.0 | 1.3 | 1.3 | 0.9 | |
| Variance explained by the rotated factor | 2.6 | 2.5 | 2.5 | 2.4 | 2.3 | 2.2 | |
| Factor loadings for 6 spending types and 3 stimulus checks | | | | | | | |
| Loans 1 | **0.88** | −0.02 | 0.09 | 0.06 | 0.15 | 0.11 | 0.81 |
| Loans 2 | **0.91** | 0.00 | 0.09 | 0.06 | 0.08 | 0.16 | 0.87 |
| Loans 3 | **0.85** | 0.03 | 0.15 | 0.03 | 0.07 | 0.23 | 0.81 |
| Savings 1 | −0.01 | **0.87** | 0.01 | −0.14 | 0.03 | 0.13 | 0.79 |
| Savings 2 | 0.02 | **0.93** | −0.03 | −0.03 | 0.04 | 0.07 | 0.88 |
| Savings 3 | 0.01 | **0.90** | −0.01 | −0.03 | 0.10 | 0.03 | 0.82 |
| Housing 1 | 0.13 | −0.02 | **0.78** | 0.23 | 0.21 | 0.24 | 0.78 |
| Housing 2 | 0.12 | −0.01 | **0.88** | 0.17 | 0.16 | 0.15 | 0.87 |
| Housing 3 | 0.13 | −0.01 | **0.87** | 0.20 | 0.15 | 0.13 | 0.85 |
| Household supplies 1 | 0.09 | −0.04 | 0.16 | **0.84** | 0.12 | 0.09 | 0.75 |
| Household supplies 2 | 0.01 | −0.10 | 0.21 | **0.82** | 0.22 | 0.19 | 0.82 |
| Household supplies 3 | 0.06 | −0.09 | 0.20 | **0.82** | 0.09 | 0.26 | 0.79 |
| Durable goods 1 | 0.06 | 0.04 | 0.15 | 0.09 | **0.77** | 0.20 | 0.67 |
| Durable goods 2 | 0.09 | 0.11 | 0.20 | 0.16 | **0.81** | 0.13 | 0.76 |
| Durable goods 3 | 0.15 | 0.04 | 0.12 | 0.14 | **0.81** | 0.15 | 0.74 |
| Medical costs 1 | 0.30 | 0.21 | 0.20 | 0.24 | 0.23 | **0.70** | **0.78** |
| Medical costs 2 | 0.21 | 0.08 | 0.21 | 0.17 | 0.22 | **0.83** | **0.86** |
| Medical costs 3 | 0.19 | 0.08 | 0.18 | 0.23 | 0.21 | **0.80** | **0.81** |

*Note:* Rotated factor pattern. Loadings 0.70 or larger are in bold.

**Statistical analysis.** We used multiple linear regression to identify associations of FHS-LF composite scale scores and FHS subscale scores (dependent variables) with receipt of stimulus checks and spending factors of stimulus checks (independent variables). We also assessed interactions between the family health scores of income groups (low or mid-to-high income; independent variable) and receipt of stimulus checks or spending factors. Items regarding receipt and spending of stimulus checks were used as exposure variables in each model, and demographics were used as control variables. All results were adjusted for age, SES, employment status, gender, education, race, income level, relationship status, and household size. Multivariable models were based on 290–456 participants who had complete data for each model; spending factors analyses had smaller sample size than the receipt of stimulus checks analysis. The threshold of significance was $\alpha = 0.05$. To complete this statistical analysis, we used statistical software package SAS Studio 3.8.

## Results

### Descriptive statistics

The majority of participants in this study identified as middle income (56% [52% in U.S.]; [31]), white (73%, [76% in U.S.]; [32]), female (52%), a college graduate (45%, [33% in U.S.]; [32]), employed (83%, [63% in U.S.]; [32]), and married (55%, [50% in U.S.]; [33]). Additionally, the mean age among participants was 40 years old, the mean household size was 3 (2.6 in U.S.; [32]), and the mean subjective socioeconomic status score was 5.2 out of 10 (Table 2).

### Stimulus check analysis

In Table 3, we present mean scores on family health measures for groups of participants who received less than 3 checks and those who received 3 checks, low income and mid-to-high income groups, and differences in the mean family health scores across those groups. For example, after adjustment, the mean FHS score among individuals who received all three checks was 91.8 points (95% CI = [89.6, 94.0]), which was 8.0 points higher (95% CI = [2.1, 13.8]) than the mean FHS score among those who received less than three checks (83.9 points [95% CI = (78.5, 89.3)]). Mean family social-emotional health scores were also higher among individuals who received three checks. However, there were no significant differences for the individual subscales of family healthy lifestyle, family health resource, and family external social support, although borderline. Associations of receiving all three checks with FHS score and subscale scores did not differ significantly by income level (Table 3).

### Spending factor analysis

Mean differences of FHS composite and subscale scores varied across the spending factors (Fig 1 leftmost panel and S2 Tables). After adjustment, mean FHS scores were significantly lower among individuals who spent greater portions of their stimulus checks on housing, household supplies, and medical costs. For example, mean FHS scores were 4.7 points lower (95% CI = [−7.0, −2.3]) per standard deviation higher on the housing spending profile. Mean family social-emotional health scores were also significantly lower among individuals who spent greater portions of their stimulus checks on housing, household supplies, and medical costs. Mean family healthy lifestyle scores were significantly higher among individuals who spent greater portions of their stimulus checks on loans. Mean family health resource scores were significantly lower among individuals who spent greater portions on loans, housing, household supplies, durable goods, and medical costs. Mean family external social support scores were significantly lower among individuals who spent greater portions of their checks on medical costs. Spending greater portions of checks on medical costs was the most consistent lower score among every measure except family healthy lifestyle.

### Income analysis

We further examined whether income groups and spending factors interacted in their associations with FHS scores through testing models that included interactions between income groups and spending factors (Fig 1 and S2 Tables).

**Table 2. Demographics and Family Health Scale Mean Scores.**

| | Total sample | Received fewer than 3 checks | Received 3 checks | Low income | Mid-to-high income |
|---|---|---|---|---|---|
| Sample size | 456 | 72 | 384 | 148 | 273 |
| Age (mean, SD) | 39.5 (13.3) | 35.6 (13.4) | 40.2 (13.2) | 42.4 (15.2) | 38.6 (12.3) |
| **Gender** | | | | | |
| Female | 51.8 | 51.4 | 51.8 | 54.7 | 51.3 |
| Male | 47.8 | 48.6 | 47.7 | 44.6 | 48.4 |
| Self-identified | 0.4 | 0.0 | 0.5 | 0.7 | 0.4 |
| **Relationship status** | | | | | |
| Cohabiting | 14.9 | 15.3 | 14.8 | 21.0 | 12.1 |
| Married | 54.8 | 48.6 | 56.0 | 31.1 | 72.9 |
| Other | 30.3 | 36.1 | 29.2 | 48.0 | 15.0 |
| Has children | 68.4 | 63.9 | 69.3 | 54.1 | 81.0 |
| Household size (mean, SD) | 3.1 (1.4) | 3.3 (1.4) | 3 (1.4) | 2.6 (1.3) | 3.4 (1.3) |
| **Income level** | | | | | |
| Low | 32.5 | 27.8 | 33.3 | 100.0 | 0.0 |
| Middle | 55.9 | 52.8 | 56.5 | 0.0 | 93.4 |
| High | 4.0 | 8.3 | 3.1 | 0.0 | 6.6 |
| **Race** | | | | | |
| Asian | 6.4 | 6.9 | 6.3 | 5.4 | 5.9 |
| Black/African American | 11.6 | 15.3 | 10.9 | 11.5 | 11.7 |
| Hispanic/Latinx | 5.0 | 5.6 | 5.0 | 0.7 | 7.0 |
| Native American | 1.1 | 1.4 | 1.0 | 0.7 | 1.5 |
| Multiracial | 2.9 | 5.6 | 2.3 | 4.7 | 1.5 |
| White/Caucasian | 72.8 | 65.3 | 74.2 | 77.0 | 72.5 |
| **Socioeconomic status** (mean, SD)* | 5.2 (2.2) | 5.5 (2) | 5.2 (2.3) | 4.1 (2) | 6.1(1.9) |
| **Education** | | | | | |
| Less than high school | 1.1 | 1.4 | 1.0 | 2.0 | 0.0 |
| High school graduate | 10.8 | 6.9 | 11.5 | 15.5 | 6.6 |
| Some college | 16.0 | 16.7 | 15.9 | 20.3 | 11.7 |
| 2-year degree | 11.0 | 12.5 | 10.7 | 19.6 | 6.2 |
| 4-year degree | 45.0 | 45.8 | 44.8 | 34.5 | 53.1 |
| Master's degree | 14.9 | 16.7 | 14.6 | 6.1 | 21.3 |
| Professional or doctoral degree | 1.3 | 0.0 | 1.6 | 2.0 | 1.1 |
| **Employment status** | | | | | |
| Employed | 83.1 | 86.1 | 82.6 | 75.7 | 91.9 |
| Unemployed | 16.0 | 12.5 | 16.7 | 24.3 | 7.0 |
| **Stimulus checks received** | | | | | |
| 0 | 3.5 | 22.2 | 0.0 | 3.4 | 3.7 |
| 1 | 3.7 | 23.6 | 0.0 | 3.4 | 2.9 |
| 2 | 8.6 | 54.2 | 0.0 | 6.8 | 9.5 |
| 3 | 84.2 | 0.0 | 100.0 | 86.5 | 83.9 |
| **Mean scale scores** | | | | | |
| FHS composite score (Max = 128) | 88.8 (21.4) | 83.4 (18.8) | 89.8 (21.8) | 84.8 (20.2) | 92.5 (20.7) |
| Family social-emotional health (Max = 52) | 39.5 (10.3) | 36.6 (11.1) | 40.1 (10.1) | 38.4 (10.5) | 40.9 (9.1) |
| Family healthy lifestyle (Max = 24) | 17.1 (4.7) | 16.6 (4.9) | 17.2 (4.7) | 15.5 (4.9) | 18.4 (3.8) |
| Family health resources (Max = 36) | 22.0 (9.7) | 21.0 (8.3) | 22.2 (9.9) | 21.9 (8.4) | 21.8 (10.6) |
| Family external social supports (Max = 16) | 10.2 (4.4) | 9.3 (3.9) | 10.3 (4.5) | 9.0 (4.5) | 11.3 (3.8) |

Reported percentages are column percentages.

*[34]

**Table 3. Associations of stimulus check receipt and income level with mean scores of family health measures.**

| Checks Received | Overall | | Low income | | Mid-to-high income | | Mean difference (low income - mid-to-high income) | |
|---|---|---|---|---|---|---|---|---|
| | Mean | 95% CI | Mean | 95% CI | Mean | 95% CI | Mean | 95% CI |
| **FHS composite** | | | | | | | | |
| Less than 3 checks received | 83.9 | (78.5, 89.3) | 76.0 | (65.5, 86.5) | 88.4 | (82.0, 94.9) | −12.4 | (−25.0, 0.14) |
| 3 checks received | 91.8 | (89.6, 94.0) | 83.1 | (79.0, 87.1) | 96.8 | (93.7, 99.8) | −13.7 | (−19.1, −8.3) |
| Mean difference (all 3 − less than 3) | 8.0 | (2.1, 13.8) | 7.1 | (−4.0, 18.1) | 8.3 | (1.4, 15.2) | −1.3 | (−14.3, 11.8) |
| P-value for difference | | 0.007* | | 0.209 | | 0.018* | | 0.850 |
| **Social-emotional health** | | | | | | | | |
| Less than 3 checks received | 36.7 | (34.2, 39.2) | 32.1 | (27.1, 37.0) | 38.9 | (35.9, 41.9) | −6.9 | (−12.8, −1.0) |
| 3 checks received | 41.0 | (39.9, 42.0) | 38.7 | (36.9, 40.6) | 42.2 | (40.8, 43.6) | −3.5 | (−6.0, −0.9) |
| Mean difference (all 3 − less than 3 | 4.2 | (1.5, 7.0) | 6.7 | (1.6, 11.8) | 3.3 | (0.0, 6.5) | 3.4 | (−2.7, 9.5) |
| P-value for difference | | 0.002* | | 0.011* | | 0.047* | | 0.272 |
| **Healthy lifestyle** | | | | | | | | |
| Less than 3 checks received | 16.5 | (15.4, 17.6) | 15.4 | (13.3, 17.6) | 17.2 | (15.8, 18.5) | −1.8 | (−4.3, 0.8) |
| 3 checks received | 17.4 | (16.9, 17.8) | 16.1 | (15.3, 17.0) | 18.1 | (17.5, 18.7) | −2.0 | (−3.1, −0.8) |
| Mean difference (all 3 − less than 3) | 0.9 | (−0.3, 2.1) | 0.7 | (−1.6, 3.0) | 0.9 | (−0.5, 2.3) | −0.2 | (−2.9, 2.5) |
| P-value for difference | | 0.161 | | 0.537 | | 0.208 | | 0.884 |
| **Health resources** | | | | | | | | |
| Less than 3 checks received | 21.1 | (18.7, 23.4) | 19.7 | (15.2, 24.3) | 22.3 | (19.5, 25.1) | −2.6 | (−8.0, 2.8) |
| 3 checks received | 22.8 | (21.9, 23.8) | 18.5 | (16.8, 20.3) | 25.3 | (24.0, 26.6) | −6.8 | (−9.1, −4.4) |
| Mean difference (all 3 − less than 3) | 1.8 | (−0.8, 4.3) | −1.2 | (−6.0, 3.6) | 3.0 | (−0.0, 5.9) | −4.2 | (−9.8, 1.5) |
| P-value for difference | | 0.169 | | 0.620 | | 0.052 | | 0.149 |
| **Social support** | | | | | | | | |
| Less than 3 checks received | 9.6 | (8.5, 10.6) | 8.8 | (6.8, 10.8) | 10.0 | (8.8, 11.3) | −1.2 | (−3.6, 1.2) |
| 3 checks received | 10.6 | (10.2, 11.1) | 9.7 | (8.9, 10.4) | 11.2 | (10.6, 11.8) | −1.5 | (−2.6, −0.5) |
| Mean difference (all 3 − less than 3) | 1.1 | (−0.0, 2.2) | 0.9 | (−1.3, 3.0) | 1.2 | (−0.2, 2.5) | −0.3 | (−2.9, 2.2) |
| P-value for difference | | 0.057 | | 0.431 | | 0.081 | | 0.795 |

*Statistically significant difference at α = 0.05.

Means and mean differences adjusted for age, SES, employment status, gender, education, race, income level, relationship status, and household size.

**Low-income.** While low-income FHS scores were not significantly different according to spending profiles, scores were lower with spending greater portions on savings, housing, household supplies, and medical costs, and higher scores for loans and durable goods. For the subscales, as spending increased, social-emotional health scores were significantly lower on medical costs (mean difference: −1.8; 95% CI: [−3.5, −0.1]) and savings (mean difference: −2.0; 95% CI: [−3.7, −0.3]). Healthy lifestyle scores were significantly higher with spending greater portions on loans (mean difference: 1.7; 95% CI: [0.9, 2.4]). Resource scores significantly decreased among the low-income group with spending greater portions on housing (mean difference: −1.8; 95% CI: [−3.3, −0.4]). Family external social support scores among the low-income group were not significantly different by spending profiles (Fig 1 middle panel and S2 Tables).

**Mid-to-high income.** In the mid-to-high income group, FHS scores were significantly lower with higher spending on housing (mean difference: −6.2; 95% CI: [−9.3, −3.1]), household supplies (mean difference: −4.6; 95% CI: [−7.5, −1.7]), durable goods (mean difference: −4.3; 95% CI: [−7.3, −1.3]), or medical costs (mean difference: −8.7; 95% CI: [−11.6, −5.7]). Social emotional health scores significantly decreased with increased spending on medical costs (mean difference: −2.4; 95% CI: [−3.8, −1.0]) and housing (mean difference: −1.9; 95% CI: [−3.4, −0.5]), and borderline significant on

household supplies (mean difference: −1.3; 95% CI: [−2.7, 0.0]). Resource scores significantly decreased with spending greater portions on loans (mean difference: −2.0; 95% CI: [−3.2, −0.7]), housing (mean difference: −3.3; 95% CI: [−4.6, −2.0]), household supplies (mean difference: −3.0; 95% CI: [−4.2, −1.7]), durable goods (mean difference: −3.4; 95% CI: [−4.7, −2.2]), and medical costs (mean difference: −4.9; 95% CI: [−6.1, −3.6]). Only spending more on medical costs was significantly associated with a lower external social support score (mean difference: −0.9; 95% CI: [−1.5, −0.3]), however, spending more on housing and household supplies were borderline significant (see Fig 1 and S2 Tables).

**Difference between income groups.** The association of FHS scores with spending on medical costs and durable goods were significantly different between the low and mid-to-high income groups. For example, the reduction in FHS score per SD of medical spending was 5.6 points (95% CI: [1.1, 10.2]) more pronounced among the mid-to-high income group (8.7 points lower FHS score per SD of medical spending) than among the low-income group (3.0 points lower FHS score per SD of medical spending). For the family social-emotional health subscale, the effect of spending on savings was significantly more pronounced among the low-income group (2 points lower) than the mid-to-high income group (0.2 points higher), with a reduction in scores by 2.2 points (95% CI: [−4.3, −0.1]). An increase in healthy lifestyle scores per SD of spending on loans was 1.4 points (95% CI: [0.4, 2.3]) more pronounced among the low-income group (1.6 points higher score) than the mid-to-high income group (0.3 points higher score). The reduction in resource scores per SD of spending on loans, household supplies, durable goods, and medical costs were more pronounced among the mid-to-high income group than the low-income group. For example, the reduction of scores per SD of medical spending was 4.2 points (95% CI: [2.4, 6.1]) more pronounced among the mid-to-high income group (4.9 points lower) versus the low-income group (0.6 points lower). The association of external social support scores with spending profiles did not significantly differ by income group (Fig 1 and S2 Tables).

## Discussion

This study focused on the impact of U.S. government-assisted stimulus payments on family health and stressors caused by the COVID-19 pandemic. Our work is important because a family's reaction to stress influences its capacity to maintain or re-establish stability, a premise from the family systems theory. We confirmed that stimulus payments are associated with better family health across all income levels. Although receiving three payments positively influences family health and stress levels, the association of check receipt with family health did not significantly differ by income group. Families whose spending choices reflect immediate socio-economic demands had worse family health. Lastly, stimulus spending for immediate resource needs (such as housing and medical costs) was associated with worse family health across most family health subscales. This effect was more pronounced among those in the mid-to-high income group.

### Effect of stimulus payments on family health

Our study addresses the gap in the literature regarding the impact of receiving stimulus checks on family health. Existing research has demonstrated the overall effect of the COVID-19 pandemic on family systems, such as decreased mental health, lower levels of family engagement, and reduced family cohesion [8,14,22,23,35–37]. Financial stress was also a significant contributor to the negative effect of the pandemic on families [3]. In similar economic situations, such as the 2008 recession, other research identified that individual wellbeing (affect and feelings of stress) improved with the one-time tax rebate payments [38]. Similarly, a recent study indicated that decreases in financial stress could reduce conflict within the family system and receiving government benefits alleviate financial stress. However, simply receiving stimulus checks did not equate with improvements in emotional closeness and relationship happiness [3]. Our results demonstrated better social-emotional health with receipt of stimulus checks, possibly due to the difference in measuring social-emotional health. Better family communication and support have been associated with increases in family wealth [39]; alternatively, financial insecurity such as unsecured debt is associated with child behavior issues [40]. Improved socio-economic health among families after receiving stimulus checks aligns with family systems theory and the association between better family health, social-emotional health, and receiving stimulus checks.

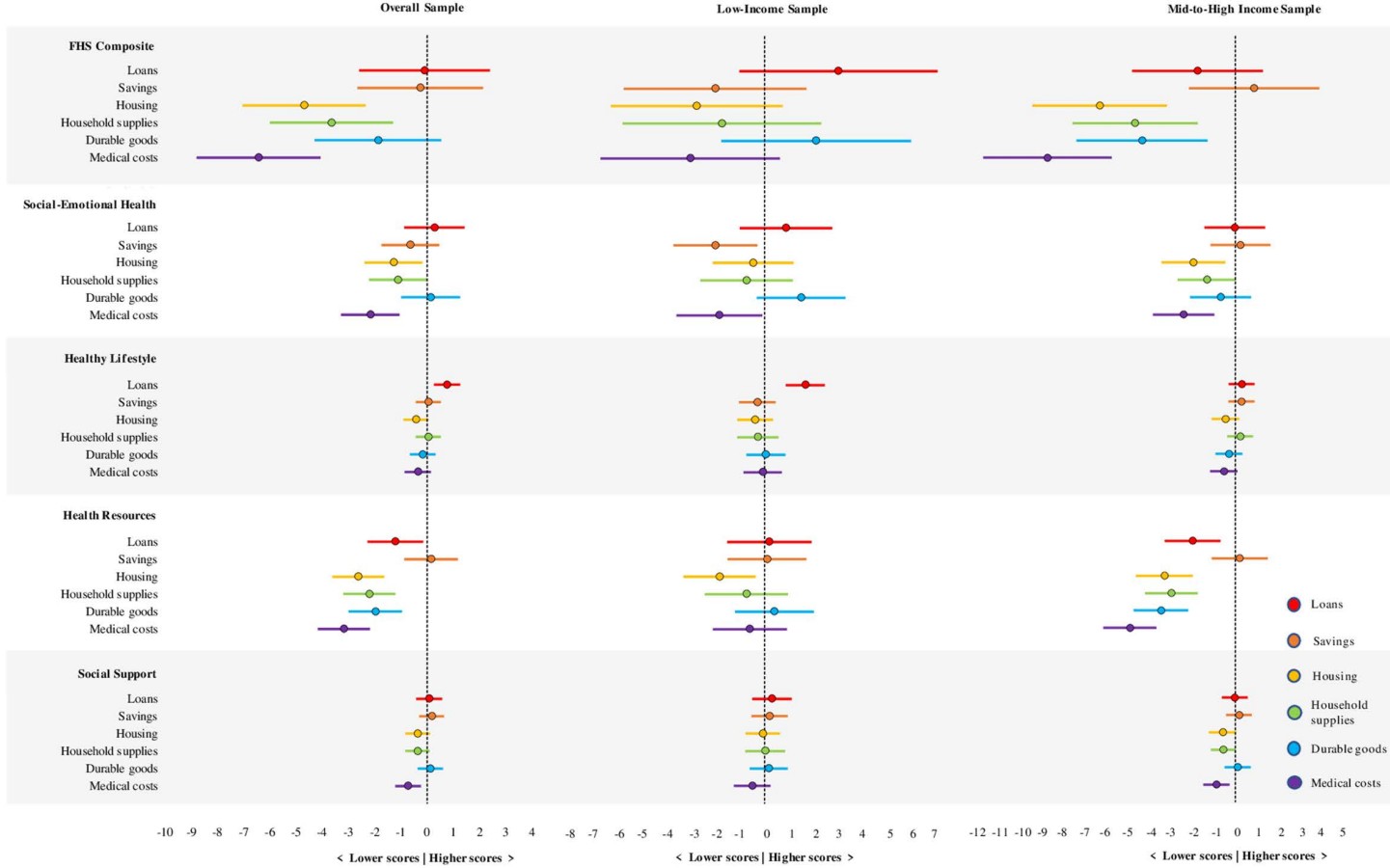

**Fig 1. Associations of six spending types with mean scores of family health measures.** The six spending types are the six factors from the principal component analysis. Each estimate plotted represents the difference in mean family health score per standard deviation increment in spending factor score, with horizontal bar showing 95% confidence interval for the difference. Estimates falling to the right of the vertical dashed line indicate that higher spending of that type is associated with higher mean family health score, and estimates falling to the left of the vertical dashed line indicate that higher spending of that type is associated with lower mean family health score. Results are shown for the entire sample of participants who received three checks (overall), and separately for low-income and mid-to-high income participants.

Within each income group (low and mid-to-high income), receiving checks was associated with better social-emotional health in families. However, the stimulus checks were not more effective for low-income families than for mid-to-higher-income families. This result seems contrary to existing research, which has found that low-income families were considered at the highest risk for the economic and situational repercussions of the COVID-19 pandemic and, therefore, most likely to benefit from receiving three stimulus checks [14,35,41,42]. While stimulus checks provide some overall benefit, our findings point to cumulative family income as more predictive of family wellbeing among lower-income families. Families with limited income entered the pandemic at a more significant disadvantage than higher-income families and were disproportionately affected [43,44]. One analysis found the first round of stimulus (CARES Act) did not provide a large enough benefit to the poor, young, and those with children [45], demonstrating that lower SES families likely began the pandemic with recurring struggles. Many low-income families worked in service industries and faced unemployment or were deemed essential workers, thus increasing their exposure to COVID-19. Some researchers found that lower SES families were more likely to report perceived harms in family physical health, family mental health, family relationships,

and decreased family income, while higher SES families were more likely to report perceived benefits in family physical health, family mental health, and family relationships [35]. Yet, higher-income individuals may have experienced more significant decreases in life satisfaction when COVID-19 began than individuals of lower-income levels [46], which could have influenced their reporting of family health in this study and reduced the difference in the effect of receiving stimulus checks. Though low-income benefits are especially noted, aid for all income levels may be necessary to improve family health in times of economic distress.

### Stimulus spending and family health

Stimulus payments offered a critical policy to mitigate spending declines [47]. Payments also provide opportunities for individuals and families to increase spending across various services (e.g., medical), economic capacity (e.g., loans and savings), and other essential categories such as housing, durable goods, and household supplies [47]. Descriptive data from the U.S. Bureau of Labor Statistics identified that low-income recipients, including ethnic minorities, often used their stimulus checks to replace regular sources of income to meet immediate needs, defined as regular expenses such as food, shelter, utilities, or household items. In contrast, households with higher gross income increasingly added to savings and other long-term purchasing choices [48]. Previous research on the spending of COVID-19 stimulus payments identified that expenditures for short-term debt or immediate needs were larger than in previous economic stimulus programs in 2001 and 2008 [1]. Additionally, the relative benefit of spending on loans was positive for low-income families. We presume this positive difference favoring low-income compared to mid-to-high income families is due to the unexpected ability of low-income families to pay down a loan balance faster than making a minimum loan payment. On the other hand, families spending their stimulus dollars for medical costs or other immediate obligations may reflect having a lower income, unemployment, underemployment, job loss or disruption, or illness, including contracting COVID-19.

Paying for medical expenses was the most common significant result in the spending analysis for worsened family health overall. Spending more significant portions of stimulus checks on medical costs was also associated with worse family social-emotional health, fewer family health resources, and less social support. These results are likely because a) one family member's expenses may adversely affect the whole family, as identified by family systems theory, b) growing medical bills make families financially vulnerable to pay for other essential needs such as food, clothing, housing, or other essential needs, and c) changes in jobs or insurance coverage may put families in a compromised position and lead to feelings of vulnerability and stress [49]. Additionally, spending greater portions on medical costs during the pandemic, a time when medical services were limited, may have affected the income groups differently. Our results demonstrate that the association of family health scores with spending on medical costs significantly differed between income groups. Low-income families may have already struggled financially before the pandemic and experienced overall worse family health, thus experiencing a smaller decrease in family health with increased spending on medical costs. The mid-to-high income group had significantly lower scores with increased spending on medical costs, possibly because of unexpected or acute financial stresses. For example, when medical expenses rose among mid-to-high income families, they may have experienced more limited-service access and higher medical costs than normal, decreasing family health scores. Lower income families may experience ongoing medical expenses and already limited access to medical services due to unique barriers including complications in accessing or receiving insurance, distrust of healthcare providers, and lack of education [50].

The effect of receiving stimulus checks seems to be most beneficial for family health among mid-to-high income families. Those without savings before the pandemic may have used the stimulus money to pay immediate living expenses, while those with savings balances were more likely to experience gains in savings (white middle class, well employed) and pay for less-essential items [1]. Despite the expected benefit from receiving stimulus checks, family health scores in our study were still the lowest among the mid-to-high income group when stimulus checks were spent on items for immediate needs when compared to scores for low-income families. It is possible that some mid-to-high income families

unexpectedly experienced decreases in income or faced changes in employment, for example, thus prompting them to use their checks for immediate needs and loan balances. Other research similarly identified more significant wellbeing decreases among higher SES families when COVID-19 began [46]. Mid-to-high income families may have experienced a higher magnitude of loss than low-income families during the pandemic. However, low-income families still maintained lower family health scores overall, and in all subscales (see S2 Tables).

### Limitations

First, like other studies, the cross-sectional sample drawn from Amazon's MTurk population may not represent the broader U.S. population or those who received some or all of the stimulus checks during COVID-19 [51]. However, stratified sampling for target percentages was used to approximate population markers such as the percentage of low-income participants, marital status, and other factors to help compensate for the cross-sectional sample. Second, the sample size reduced statistical power to detect significance. The non-significant results may become more or less apparent through future studies involving larger samples. Third, respondents may have interpreted savings as either setting aside cash, depositing money in an existing savings account, or investing in long-term savings vehicles for retirement or related purposes. As a result, our understanding of savings' role in stimulus checks may have been limited. Future studies should better specify between these savings options since each purpose is distinct. Fourth, the survey did not include questions about overall monthly or daily spending habits, which would have provided additional context to understand reasons for differences in stimulus check spending. However, we adjusted our models for income, which provides partial information about participants' financial circumstances. Last, insignificant scores in the subscales may have been due to a lower item count within each subscale. For example, the social-emotional health subscale had the highest item count (apart from the overall FHS composite score) compared to the external social support subscale, which had only 16 items.

### Implications and conclusion

Stimulus payments may be a promising family policy method for improving overall family health and well-being during challenging circumstances. Our study provided insight into areas of expenditure which may have affected family health more than others, namely medical expenses. Worse family health with spending on medical expenses may reflect a larger issue in the U.S. healthcare system. Low-income families experience worse family health overall compared to mid-to-high income families. Since receiving stimulus checks improved family health among mid-to-high income families, it is possible that the structure and dissemination of stimulus checks was more favorable for mid-to-high income families rather than low-income families.

Future research should identify the differences between income groups in receiving government aid or test a variety of government aid methods. Such research may include further analysis of nontraditional family structures within income groups or consider cost of living in various locations. The U.S. government could prioritize more geo- and category-targeted methods that benefit strategic needs among low-income families and consider the cost of living in their location. Regardless, family-level health benefits are essential for policymakers and resource distribution planners to consider for future stimulus payment needs.

### Supporting information

**S1 Table. Spending variable correlation matrix.**
(DOCX)

**S2 Tables. Association of six spending types with mean family health scores.**
(DOCX)

**S1 Data. Dataset for analysis.**
(CSV)

## Acknowledgments

Maddison Dillon, MPH, was involved in the beginning stages of the research and contributed to survey methods and data cleaning.

## Author contributions

**Conceptualization:** Emma M. Reese, Noah Lines, Evan L. Thacker, Michael D. Barnes.

**Data curation:** Emma M. Reese, Noah Lines, Evan L. Thacker.

**Formal analysis:** Emma M. Reese, Evan L. Thacker.

**Funding acquisition:** Michael D. Barnes.

**Investigation:** Emma M. Reese, Noah Lines, Michael D. Barnes.

**Methodology:** Emma M. Reese, Noah Lines, Evan L. Thacker, Michael D. Barnes.

**Project administration:** Emma M. Reese, Michael D. Barnes.

**Resources:** Michael D. Barnes.

**Software:** Emma M. Reese, Evan L. Thacker.

**Supervision:** Michael D. Barnes.

**Validation:** Emma M. Reese.

**Visualization:** Emma M. Reese, Evan L. Thacker.

**Writing – original draft:** Emma M. Reese, Noah Lines, Michael D. Barnes.

**Writing – review & editing:** Emma M. Reese, Noah Lines, Evan L. Thacker, Michael D. Barnes.

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
