## [Decision Letter · Decision Letter 0]

3 Nov 2023

Dear Dr. Reese,

We look forward to receiving your revised manuscript.

Kind regards,

Maurizio Fiaschetti

Academic Editor

PLOS ONE

Journal Requirements:

2. Please ensure that you have specified (1) whether consent was informed and (2) what type you obtained (for instance, written or verbal, and if verbal, how it was documented and witnessed). If your study included minors, state whether you obtained consent from parents or guardians. If the need for consent was waived by the ethics committee, please include this information.

Reviewers' comments:

Reviewer's Responses to Questions

**Comments to the Author**

1. Is the manuscript technically sound, and do the data support the conclusions?

Reviewer #1: Partly

Reviewer #2: Yes

2. Has the statistical analysis been performed appropriately and rigorously?

Reviewer #1: No

Reviewer #2: Yes

3. Have the authors made all data underlying the findings in their manuscript fully available?

Reviewer #1: Yes

Reviewer #2: Yes

4. Is the manuscript presented in an intelligible fashion and written in standard English?

Reviewer #1: Yes

Reviewer #2: Yes

Reviewer #1: This paper examines the association between COVID-19 stimulus checks and family health status. The authors collected approximately 500 data points and studies whether receiving full three stimulus checks is linked with better family health status and the heterogeneous effect across income groups.

Major revision:

(a) The authors described the sample collected through MTurk. It would help the readers to understand the MTurk sample by adding comparison with the representative census sample.

(b) The survey asked about the stimulus check spendings. However, participants may not have a clear mental accounting of daily spending between the stimulus checks and their regular income. Thus adding a total spending question as an extra control will improve the result interpretation.

(c) The authors did not clearly explain the exploratory factor analysis process. The methods were not clearly specified. And there lacks robustness check on how the variables changes with different selection rules or pre-specified number of variables. Additionally, the authors did not show the correlation across selected variables, thus hard to judge if the models would suffer from multicollinearity problem.

(d) The authors fail to provide informative summary statistics. The paper aims to understand the difference between participants that receive three stimulus checks and those who did not receive all three. But the authors did not provide the difference of these two groups in the descriptive statistics section.

(e) The main models in the paper use whether three checks received as the key varibles. It is worthy to discretize the key variable of interest into groups that receive one/two/three checks.

Minor revision:

(a) The paper did literature review in the introduction section. However, the authors did not fully specify how the paper relates to the existing literature until the final discussion. It would be helpful to make it clear in the introduction part.

(b) Tables lack enough explanations. The authors should add more table notes.

Reviewer #2: The authors aimed to determine the impact of U.S. government stimulus payments on family health during the COVID-19 the pandemic. This study can help to understand the impact of government assistance on overall family health. Considering the limitations mentioned by the authors, the study is well designed and the manuscript is by and large well developed. Here are just a few structural flaws, which are as follows:

1- It is suggested that in the background part of the Abstract, the objectives of the study should be expressed in the form of a brief statement rather than in the form of numbered hypotheses.

2- The introduction section is very long. Most of the content provided is related to the discussion section. It is necessary to review and summarize this part.

3- According to the guidelines of the journal, it is suggested that the objectives of the study should be expressed in a brief statement at the end of the Introduction section.

4- The final conclusion of the study should only be developed based on the main results of the study and content from other studies should not be presented and cited. Therefore, this section should be revised based on the results of the study and their implications.

**Do you want your identity to be public for this peer review?** For information about this choice, including consent withdrawal, please see our Privacy Policy

Reviewer #1: No

Reviewer #2: **Yes: ** Yaser Sarikhani

---

## [Author Response · Author response to Decision Letter 1]

16 Dec 2023

We would like to thank the reviewers for their suggestions. We have addressed and responded to each of the issues raised below.

Reviewer #1

Major revision:

(a) The authors described the sample collected through MTurk. It would help the readers to understand the MTurk sample by adding comparison with the representative census sample.

- We appreciate the suggestion to describe the collection sample because it helps to clarify our sampling procedure through mTurk. We rewrote the sentence describing the mTurk sample in the Participants and Sampling paragraph to reflect better why and how the mTurk selection filters were created. We removed 'representative' because it suggests we sampled according to a representative census sample. We emphasize that our sample is aimed at specific qualification criteria, including marital status and lower income thresholds since those participants were most likely to qualify for the three stimulus checks.

(b) The survey asked about the stimulus check spendings. However, participants may not have a clear mental accounting of daily spending between the stimulus checks and their regular income. Thus adding a total spending question as an extra control will improve the result interpretation.

- We thank the reviewer for raising this question about the context of overall spending. The survey did not include a total spending question. We have added the following sentences to the Limitations paragraph in the Discussion section acknowledging this limitation: "The survey did not include questions about overall monthly or daily spending habits, which would have provided additional context to understand reasons for differences in stimulus check spending. However, we adjusted our models for income, which provides partial information about participants’ financial circumstances.”

(c) The authors did not clearly explain the exploratory factor analysis process. The methods were not clearly specified. And there lacks robustness check on how the variables changes with different selection rules or pre-specified number of variables. Additionally, the authors did not show the correlation across selected variables, thus hard to judge if the models would suffer from multicollinearity problem.

- We thank the reviewer for inviting us to explain the factor analysis more clearly. We made extensive revisions to the Spending Patterns Determination paragraph in the Methods section, not quoted here but shown in track changes in the manuscript. We also added information about the principal component analysis to Table 1, including Eigenvalues, percent variance explained, and communality estimates. Finally, we added a Supplemental Table containing the correlation matrix for the 18 variables that were considered in the principal component analysis.

(d) The authors fail to provide informative summary statistics. The paper aims to understand the difference between participants that receive three stimulus checks and those who did not receive all three. But the authors did not provide the difference of these two groups in the descriptive statistics section.

- We thank the reviewer for suggesting that we display descriptive statistics for the group of participants who received three stimulus checks and the group that received fewer than three stimulus checks. We have added two columns to Table 2 to display descriptive statistics for those groups.

(e) The main models in the paper use whether three checks received as the key variables. It is worthy to discretize the key variable of interest into groups that receive one/two/three checks.

- We thank the reviewer for suggesting that we analyze separately the groups who received only two checks or only one check. For our analyses, we had decided to dichotomize the number of checks received to the categories "three" vs "fewer than three" because of small sample sizes for the groups of survey respondents that received only two, only one, or no stimulus checks. We have added to Table 2 the percentages of participants who received zero (3.5%), one (3.7%), two (8.6%), or three (84.2%) stimulus checks. We have also added to the Stimulus Check paragraph of the Methods section the following explanation: "For analyses based on the number of checks received as an independent variable, we dichotomized the responses to “three checks” versus “fewer than three checks” because of relatively small sample sizes for groups that reported receiving only two, only one, or no stimulus checks."

Minor revision:

(a) The paper did literature review in the introduction section. However, the authors did not fully specify how the paper relates to the existing literature until the final discussion. It would be helpful to make it clear in the introduction part.

- We thank the reviewer for suggesting we improve the introduction. We shortened the introduction to add clarity and make it more readable and shifted essential discussion-oriented elements to that section. This reviewer's observation helps improve both the introduction and discussion.

(b) Tables lack enough explanations. The authors should add more table notes.

- We thank the reviewer for suggesting to add more description to the tables. We have added notes to Table 3, and added more descriptive table titles, column headings, and row headings. We added a sentence at the beginning of the Stimulus Check Analysis paragraph in the Results section to clarify the table further: "In Table 3, we present mean scores on family health measures for groups of participants who received less than 3 checks and those who received 3 checks, low income and mid-to-high income groups, and differences in the mean family health scores across those groups." We have also added a Figure 1 Legend after the first reference to Figure 1 in the text which gives a more detailed description of the figure.

Reviewer #2

1- It is suggested that in the background part of the Abstract, the objectives of the study should be expressed in the form of a brief statement rather than in the form of numbered hypotheses.

- We thank the reviewer for the suggestion of simplifying our hypothesis into a brief statement. We have done so in both the Abstract and at the end of the Introduction (Purpose section). Our objective statement is as follows: "We hypothesized that receiving stimulus checks is associated with better family health and the effect of stimulus check receipt differs by income level. Additionally, we hypothesized that spending on immediate needs and paying off loans is associated with worse family health and the effects differ by income level." Further, we have removed reference to hypothesis numbers and replaced those references with more specific text regarding the objective statement in the Abstract, Introduction, and first paragraph of the Discussion section.

2- The introduction section is very long. Most of the content provided is related to the discussion section. It is necessary to review and summarize this part.

- We thank the reviewer for suggesting we shorten the introduction. We shortened the introduction to make it more readable and shifted essential discussion-oriented elements to that section. This reviewer's observation helps improve both the introduction and discussion.

3- According to the guidelines of the journal, it is suggested that the objectives of the study should be expressed in a brief statement at the end of the Introduction section.

- We thank the reviewer for the suggestion to add our hypothesis at the end of the Introduction (Purpose section) as a brief statement. We have done so and it appears as follows: "We hypothesized that receiving stimulus checks is associated with better family health and the effect of stimulus check receipt differs by income level. Additionally, we hypothesized that spending on immediate needs and paying off loans is associated with worse family health and the effects differ by income level."

4- The final conclusion of the study should only be developed based on the main results of the study and content from other studies should not be presented and cited. Therefore, this section should be revised based on the results of the study and their implications.

- We thank the reviewer for the suggestion to adjust the conclusion. We have made changes to the conclusion, including removing citations and content from other studies. Please see the Implications and Conclusion section for those changes.

---

## [Decision Letter · Decision Letter 1]

4 Nov 2024

Dear Dr. Reese,

Thank you for submitting your manuscript to PLOS ONE. After careful consideration, we feel that it has merit but does not fully meet PLOS ONE’s publication criteria as it currently stands. Therefore, we invite you to submit a revised version of the manuscript that addresses the points raised during the review process.

We look forward to receiving your revised manuscript.

Kind regards,

Maurizio Fiaschetti

Academic Editor

PLOS ONE

Journal Requirements:

Reviewers' comments:

Reviewer's Responses to Questions

**Comments to the Author**

Reviewer #2: (No Response)

Reviewer #3: (No Response)

Reviewer #4: (No Response)

2. Is the manuscript technically sound, and do the data support the conclusions?

Reviewer #2: Yes

Reviewer #3: Yes

Reviewer #4: Partly

3. Has the statistical analysis been performed appropriately and rigorously?

Reviewer #2: Yes

Reviewer #3: Yes

Reviewer #4: Yes

4. Have the authors made all data underlying the findings in their manuscript fully available?

Reviewer #2: Yes

Reviewer #3: Yes

Reviewer #4: Yes

5. Is the manuscript presented in an intelligible fashion and written in standard English?

Reviewer #2: Yes

Reviewer #3: Yes

Reviewer #4: Yes

Reviewer #2: Considering the changes made in the text based on the comments of the first round of review, it seems that the authors have put enough effort to improve the the manuscript. Therefore, there is no more comment.

Reviewer #3: 1. Perhaps I did not see, or my ignorance, but mTurk is unfamiliar to me ...if possible pls. add something from "Amazon Mechanical Turk (MTurk) is a crowdsourcing marketplace that makes it easier for individuals and businesses to outsource their processes and jobs to a distributed workforce who can perform these tasks virtually. This could include anything from conducting simple data validation and research to more subjective tasks like survey participation, content moderation, and more. MTurk enables companies to harness the collective intelligence, skills, and insights from a global workforce to streamline business processes, augment data collection and analysis, and accelerate machine learning development."

2."147 known as active response rate [2432]. The 500-participant sample size would allow sufficient

1" I see 456=n, please fix this minor item.

3. The paper defines income categories differently in various sections. For example, it mentions "<$25,000" as low income in one part, but later uses "<$40,000" as the low-income threshold. or perhaps it is on the cut-off, if possible please make it clearer.

4. The explanation of factor analysis mentions six factors, but Eigenvalues only exceed 1.0 for five factors. The text acknowledges this discrepancy without resolving it, raising doubts about whether the six-factor solution is valid (lines 226-229). A non-zero vector v is an eigenvector of A if Av = λv for some number λ, called the corresponding eigenvalue. The eigenvector with the largest eigenvalue is the direction with most variability, this eigenvector is the first principle component. It may be possible there is some factor instability drift and heteroscedasticity (which is fine), "Agrrawal and Clark. "ETF Betas: A Study of their Estimation Sensitivity to Varying Time Intervals." ETFs and Indexing (2007)", attribute factor/coefficient instability to heterogenous variance discontinuities and note their impact on multiple orthogonal factors for an overall ranking scale. A similar effect could slightly be at play here. Additionally data driven methods to identify spatial patterns by optimizing euclidean distances is utilized in "Heumann et al. Data-Driven Algorithm to Redefine the US Rural Landscape: Affinity Propagation as a Mixed-Data/Mixed-Method Tool. Economic Development Quarterly (2022)."

5. "Figure 1 Legend" section, where the explanation appears without proper figure presentation/title (lines 308-317), an appendix perhaps?

6. [line 84-86] Most U.S. research points to economic and employment

constraints [10-12-14], interruptions to family routines [135], reduced quality of life and family well-being [810, 146], psychological distress [810, 157-179]. A recent large scale study using 2 million non-natural deaths using CDC and NVDRS data establishes a lagged link between finance induced stress and subsequent period suicides. (Sandweiss et al. "Suicides as a response to adverse market sentiment. PLoS One, 2017."

A very good and robustly researched paper, a few clarifications and edits would increase its clarity and visibility. Best.

Reviewer #4: Your data sources are complex, it may be difficult for researcher students to provide such data. You can mention an early stage AI knowledge paper that shows that the utilization of automated web-harvesting algorithms can easily provide the researcher with zero-cost machine readable datasets for further analysis.

Some ambiguity wrt your eigenvector decomposition < 1 seems to exist. PCA modeling could be tightened.

**Do you want your identity to be public for this peer review?** For information about this choice, including consent withdrawal, please see our Privacy Policy

Reviewer #2: **Yes: ** Yaser Sarikhani

Reviewer #3: No

Reviewer #4: No

---

## [Author Response · Author response to Decision Letter 2]

22 Apr 2025

Response to Reviewers

We would like to thank the reviewers for their suggestions. We have addressed and responded to each of the issues raised below.

Reviewer #3

1. Perhaps I did not see, or my ignorance, but mTurk is unfamiliar to me ...if possible pls. add something from "Amazon Mechanical Turk (MTurk) is a crowdsourcing marketplace that makes it easier for individuals and businesses to outsource their processes and jobs to a distributed workforce who can perform these tasks virtually. This could include anything from conducting simple data validation and research to more subjective tasks like survey participation, content moderation, and more. MTurk enables companies to harness the collective intelligence, skills, and insights from a global workforce to streamline business processes, augment data collection and analysis, and accelerate machine learning development."

We appreciate the suggestion to elaborate on Amazon mTurk. We have added/revised some sentences in the “Participants and Sampling” paragraph to further explain how mTurk works: "MTurk is a crowdsourcing marketplace that businesses and individuals can use to outsource data collection and processing for various purposes, including research. MTurk users provide a sample more diverse than typical convenience samples, and complete tasks requested by researchers, such as surveys, data validation, and more. The mTurk users receive an incentive, typically financial compensation, for their work."

2."147 known as active response rate [2432]. The 500-participant sample size would allow sufficient 1" I see 456=n, please fix this minor item.

We thank the reviewer for identifying this mistake. We have changed the manuscript to read, "The 456-participant sample size..."

3. The paper defines income categories differently in various sections. For example, it mentions "<$25,000" as low income in one part, but later uses "<$40,000" as the low-income threshold. or perhaps it is on the cut-off, if possible please make it clearer.

We appreciate the reviewer identifying this discrepancy. The <$25,000 threshold mentioned in the paragraph on “Participants and Sampling” was used to ensure that the sample we gathered would include an adequate number of low income families. In contrast, the <$40,000 threshold mentioned in the paragraph on “Demographics” was used to define a low income category for the data analysis. We have revised both paragraphs to clarify this difference.

4. The explanation of factor analysis mentions six factors, but Eigenvalues only exceed 1.0 for five factors. The text acknowledges this discrepancy without resolving it, raising doubts about whether the six-factor solution is valid (lines 226-229). A non-zero vector v is an eigenvector of A if Av = λv for some number λ, called the corresponding eigenvalue. The eigenvector with the largest eigenvalue is the direction with most variability, this eigenvector is the first principle component. It may be possible there is some factor instability drift and heteroscedasticity (which is fine), "Agrrawal and Clark. "ETF Betas: A Study of their Estimation Sensitivity to Varying Time Intervals." ETFs and Indexing (2007)", attribute factor/coefficient instability to heterogenous variance discontinuities and note their impact on multiple orthogonal factors for an overall ranking scale. A similar effect could slightly be at play here. Additionally data driven methods to identify spatial patterns by optimizing euclidean distances is utilized in "Heumann et al. Data-Driven Algorithm to Redefine the US Rural Landscape: Affinity Propagation as a Mixed-Data/Mixed-Method Tool. Economic Development Quarterly (2022)."

We thank the reviewer for suggesting that we clarify our rationale for retaining six factors when the sixth factor had an eigenvalue of 0.9 instead of >1.0. We did not rely solely on the eigenvalue-greater-than-one rule. We have revised the "Spending Patterns Determination" paragraph to explain our evaluation of the principal component analysis in more detail, including our rationale for retaining six factors.

5. "Figure 1 Legend" section, where the explanation appears without proper figure presentation/title (lines 308-317), an appendix perhaps?

We thank the reviewer for recognizing this. Per PLOS One submission guidelines, "Figure captions are inserted immediately after the first paragraph in which the figure is cited. Figure files are uploaded separately." Figure 1 is first cited in the “Spending Factor Analysis” paragraph, therefore the Figure 1 legend is provided right after that paragraph, per journal instructions. There is an additional file that is uploaded separately from the manuscript file, labeled Fig 1.

6. [line 84-86] Most U.S. research points to economic and employment constraints [10-12-14], interruptions to family routines [135], reduced quality of life and family well-being [810, 146], psychological distress [810, 157-179]. A recent large scale study using 2 million non-natural deaths using CDC and NVDRS data establishes a lagged link between finance induced stress and subsequent period suicides. (Sandweiss et al. "Suicides as a response to adverse market sentiment. PLoS One, 2017."

We appreciate the reviewer sharing this reference. We have now cited it to support the point about psychological distress.

Reviewer #4

Your data sources are complex, it may be difficult for researcher students to provide such data. You can mention an early stage AI knowledge paper that shows that the utilization of automated web-harvesting algorithms can easily provide the researcher with zero-cost machine readable datasets for further analysis.

We appreciate the reviewer's suggestion of using automated web-harvesting algorithms. However, we do not do that in our paper; we use Amazon mTurk to outsource survey responses from a more diverse population. Thus, it may not be applicable to our paper to make reference to such web-harvesting algorithms. Rather, we have added/revised some sentences in the “Participants and Sampling” paragraph to further explain how mTurk works: "MTurk is a crowdsourcing marketplace that businesses and individuals can use to outsource data collection and processing for various purposes, including research. MTurk users provide a sample more diverse than typical convenience samples, and virtually complete tasks requested by researchers, such as surveys, data validation, and more. The mTurk users receive an incentive, typically financial compensation, for their work."

Some ambiguity wrt your eigenvector decomposition < 1 seems to exist. PCA modeling could be tightened.

We thank the reviewer for suggesting that we clarify our rationale for retaining six factors when the sixth factor had an eigenvalue of 0.9 instead of >1.0. We did not rely solely on the eigenvalue-greater-than-one rule. We have revised the "Spending Patterns Determination" paragraph to explain our evaluation of the principal component analysis in more detail, including our rationale for retaining six factors.

---

## [Editor Report · Decision Letter 2]

1 Jul 2025

Association of COVID-19 stimulus receipt and spending with family health

PONE-D-22-24977R2

Dear Dr. Reese,

We’re pleased to inform you that your manuscript has been judged scientifically suitable for publication and will be formally accepted for publication once it meets all outstanding technical requirements.

Kind regards,

Bruno Ventelou

Academic Editor

PLOS ONE
---

## [Editor Report · Acceptance letter]

PONE-D-22-24977R2

PLOS ONE

Dear Dr. Reese,

I'm pleased to inform you that your manuscript has been deemed suitable for publication in PLOS ONE. Congratulations! Your manuscript is now being handed over to our production team.

Kind regards,

on behalf of

Dr. Bruno Ventelou

Academic Editor

PLOS ONE